# Central and Peripheral Fatigue in Recreational Trail Runners: A Pilot Study

**DOI:** 10.3390/ijerph20010402

**Published:** 2022-12-27

**Authors:** Iker Muñoz-Pérez, Adrián Varela-Sanz, Carlos Lago-Fuentes, Rubén Navarro-Patón, Marcos Mecías-Calvo

**Affiliations:** 1Facultad de Ciencias de la Educación y Deporte, Universidad de Deusto, 48007 Bilbao, Spain; 2Physical and Sports Education Department, Faculty of Sport Sciences and Physical Education, University of A Coruna, 15179 A Coruna, Spain; 3Facultad de Ciencias de la Salud, Universidad Europea del Atlántico, 39011 Santander, Spain; 4Facultad de Formación del Profesorado, Universidade de Santiago de Compostela, 27001 Lugo, Spain

**Keywords:** vertical kilometer, trail running, running performance, heart rate variability, muscular fatigue, tensiomyography

## Abstract

Background: Understanding fatigue mechanisms is crucial for exercise performance. However, scientific evidence on non-invasive methods for assessing fatigue in trail running competitions is scarce, especially when vertical kilometer trail running races (VK) are considered. The main purpose of this study was to assess the autonomic nervous system (ANS) activity (i.e., central fatigue) and the state of muscle activation (i.e., peripheral fatigue) before and after a VK competition. Methods: A cross-sectional pilot study was performed. After applying inclusion/exclusion criteria, 8 recreational male trail runners (31.63 ± 7.21 yrs, 1.75 m ± 0.05 m, 70.38 ± 5.41 kg, BMI: 22.88 ± 0.48, running experience: 8.0 ± 3.63 yrs, weekly training volume: 58.75 ± 10.35 km) volunteered to participate and were assessed for both central (i.e., via heart rate variability, HRV) and peripheral (via tensiomyography, TMG) fatigue before and after a VK race. Results: After the VK, resting heart rate, RMSSD (*p* = 0.01 for both) and SDNN significantly decreased (*p* = 0.02), while the stress score and the sympathetic-parasympathetic ratio increased (*p* = 0.01 and *p* = 0.02, respectively). The TMG analyses suggest that runners already suffered peripheral fatigue before the VK and that 20–30 min are enough for muscular recovery after the race. In summary, our data suggest that participants experienced a pre-competition fatigue status. Further longitudinal studies are necessary to investigate the mechanisms underlying fatigue during trail running races, while training periodization and tapering strategies could play a key role for minimizing pre-competition fatigue status.

## 1. Introduction

Vertical Kilometer (VK) running races are a trail running modality that has gained importance in the last few years and are characterized by the great gradient (1000 m) that runners have to cover over a distance of less than 5000 m (regulation of the International Skyrunning Federation), usually performed in mountainous areas. While the main factors determining endurance running performance were exhaustively investigated in scientific literature (i.e., maximum oxygen consumption -VO_2max_-, velocity associated to VO_2max_-vVO_2max_-, lactate threshold -LT- and running economy -RE-) [1,2], the key factors affecting trail running performance were scarcely studied until recently. In this regard, studies have predominately focused on metabolic (e.g., VO_2max_, vVO_2max_, RE), biomechanical (e.g., vertical running speed, ground contact time and flight time, stride length and frequency, ground technicity) and neuromuscular (e.g., stiffness, lower-limb muscular endurance and extensor muscles maximum strength) parameters during both uphill and downhill running [3,4,5,6,7,8].

Considering the aforementioned key factors affecting trail running performance and the specific characteristics of VK competitions (i.e., runners are used to face slopes of more than 40%, while the duration of these challenges can range between 29 and 60 min), the physiological and neuromuscular demands are maximized due to the accumulated gradient in competition, since runners must displace their body upward against gravity, increasing mechanical power in a manner proportionate to slope [9]. In this regard, scientific evidence shows a time-dependent relationship for both the development of muscle damage [3,10,11,12] and altered myocardial function [13] after ultramarathon races, even if the competition is performed at low intensity [14]. Nevertheless, there is no study that determines the degree of peripheral and central fatigue after shorter, more intense (>LT) competitions, such as VK races.

Hence, knowing the degree of fatigue generated during trail running competitions is crucial to establish the optimal recovery time before applying high-demand training loads. On this point, a common approach to evaluate the acute changes in cardiac function and athletes’ readiness for training is the autonomic nervous system (ANS) activity monitoring via heart rate variability (HRV) evaluation. This method has proven to be valid and reliable to control and monitor endurance training, avoiding the development of non-functional overreaching and overtraining [15,16,17,18] by assessing the balance between the parasympathetic (PNS) and sympathetic nervous system (SNS) [19,20,21], thus allowing the establishment of optimal training conditions for supercompensation [22].

The use of HRV to assess the rest time required to restore ANS balance after competition has been reported in several studies [23,24], ranging from 1 to 3 days depending on the competition characteristics (e.g., distance, race profile, etc). However, athletes’ self-perception of full recovery after a 24-h competition can be up to 12 days [25]. This difference between subjective perception and objective evaluation of recovery (i.e., measured by ANS activation) may be influenced by muscle damage associated with peripheral fatigue and therefore, cannot be detected by HRV measurement. In this regard, simultaneous assessment of muscle fatigue and HRV, both pre- and post-competition, could be a suitable strategy to determine the degree of fatigue and the minimum time required for optimal recovery and subsequent performance.

To date, there are few studies using non-invasive methods, such as maximal voluntary contraction (MVC) determination and electrical stimulation, to assess the level of muscle fatigue after an endurance trail running competition [10,12,26]. Therefore, the use of a non-invasive method to assess muscle contractile capacity, such as tensiomyography (TMG), may be a novel strategy to analyze muscle activity before and after competition in order to establish the optimal relationship between activity and recovery [11,27,28,29,30,31,32,33].

Taking into account the benefits and practical applications of using non-invasive methods to evaluate performance variables (i.e., muscle fatigue and performance of ANS) during a competitive race, to the best of our knowledge, no studies have implemented these approaches in trail-mountain running races. For these reasons, the first objective of this study was to compare the state of activation of the ANS (via HRV measurement) before and after a VK trail running race in recreational trail runners (i.e., central fatigue). The second objective of the study was to assess the muscle fatigue caused by this type of competition in recreational trail runners (i.e., peripheral fatigue). We hypothesize that both HRV and TMG values will be negatively affected after the race (within 20–30 min after completion) when compared to those registered previous to the competition, but the magnitude of these changes will not be large.

## 2. Materials and Methods

### 2.1. Study Design

A cross-sectional study was conducted with pre- and post-competition evaluations regarding central and peripheral fatigue in a group of experienced recreational trail runners to determine the objectives of this investigation. The athletes took part in the Vertical Kilometer of Fuente Dé (2018), an uphill trail running race 2.6 km in distance and with a positive slope of 972 m to reach an altitude of ~1877 m.

### 2.2. Participants

Eleven recreational trail runners (10 men and 1 woman), with competition experience in these types of races of at least 3 years, voluntarily participated in this study.

The inclusion criteria for the present study were: (1) to complete all the records, (2) to finish the competition, and (3) not to suffer any injury or illness during the measurements. Once the inclusion criteria were applied, the final sample consisted of 8 male participants with the following characteristics (mean ± SD): age 31.63 ± 7.21 yrs, height 1.75 m ± 0.05 m, body weight 70.38 ± 5.41 kg, BMI 22.88 ± 0.48, running experience 8.0 ± 3.63 yrs; weekly training volume 58.75 ± 10.35 km.

The experimental procedures were explained in detail to all participants prior to the beginning of the study and they were free to withdraw from the study at any time. All of them signed a written informed consent form before the start of the study. The research was approved by the Ethics Committee of the Universidad Europea del Atlántico (CEI 21/2018), under the standards established in the Declaration of Helsinki.

### 2.3. Measurements

#### 2.3.1. Central Fatigue Assessment: HRV

To collect HRV data for each athlete and after a 1-min stabilization period, a 5-min measurement protocol was performed in the supine position in a dim light room with a temperature of 20–22 °C, with a relative humidity of 60–65% and after emptying their urinary bladder, as previously recommended [19,34]. During the recordings the authors encouraged participants to stay calm and not perform any movement throughout the measurements. Respiratory rate was not controlled during recording, these previous studies found only small differences between spontaneous and metronome-guided breathing on HRV variables [35].

The R-R intervals were registered using an HR band (Polar H10 band, Polar V800, Polar Electro Oy, Finland), with data downloaded using custom software (Polar Pro) and dumped into a .txt file without applying any filter for correction. Once .txt files were generated for each athlete and measurement (i.e., pre-post), these were imported into a specific software (HRV Kubios Version 3.5, Kuopio, Finland) [36] to process HRV data with artifact correction (i.e., settings: “custom” and “0.3”). The data processing configuration was carried out following the pre-established values by the Kubios software (Lambda = 500). Each R-R series were corrected by applying the medium threshold for beat correction, as suggested in the software. In this regard, the following variables were obtained for further analyses [37]: the square root of the mean of the squared differences between successive normal-to-normal intervals (RMSSD), the standard deviation of normal-to-normal intervals (SDNN), and the percentage of successive RR intervals that differ by more than 50 ms (pNN50) in the time domain. The stress score (SS), and the ratio that compares the activity of the SNS -measured by SS- vs. the activity of the PNS-measured by the variable SD1- (S/PS ratio), were obtained as non-linear measurements.

#### 2.3.2. Peripheral Fatigue Assessment: Contractile Muscle Properties

The muscular response of the rectus femoris (RF), vastus lateralis (VL), vastus medialis (VM), and gastrocnemius medialis (GM) of both legs was measured by TMG (TMG-100 System electrostimulator, TMG-BMC d.o.o., Ljubljana, Slovenia). All measurements were performed under static conditions, and with the muscle totally relaxed. The RF, VL and VM were measured with the participant in the supine position and the knee joint flexed at a 40° angle by means of a wedge cushion designed for that purpose. The GM was measured with the participant in the prone position and the knee joint bent at an angle of 15°, also through a specially-designed wedge cushion. A digital displacement transducer (Trans-Tek DC-DC; GK 40, Panoptik d.o.o. Ljubljana, Slovenia) incorporating a spring of 0.17 N·m^−1^ was used and placed perpendicular and directly on the skin at the area of maximal muscle mass of each muscle (established visually and on palpation of the muscle during a voluntary contraction), as previously described [38]. The two self-adhesive electrodes (5 × 5 cm^2^) (Compex Medical SA, Ecublens, Switzerland) were placed symmetrically to the sensor, following the arrangement of the fibers [39]. The positive electrode (anode) was placed in the proximal part and the negative (cathode) in the distal part, between 5–6 cm from the measurement point. The electrical stimulus (i.e., 1 ms) was applied with an electrostimulator (TMG-S1; Furlan Co., & Ltd., Ljubljana, Slovenia), while the intensity was varied (i.e., 50, 75 and 100 mAp). The intensity that reached the maximum response of the radial displacement of the muscle belly was selected [40]. In addition, periods of 10 s were established between consecutive measurements to minimize the possible effects of fatigue or muscle enhancement [40,41]. All measurements were performed by the same researcher, who had experience in collecting these types of measurements. None of the evaluated subjects presented discomfort during electrical stimulation. Maximal radial muscle-belly displacement (Dm); reaction or activation time (also known as time delay) between the initiation and 10% of Dm (Td); contraction time between 10 and 90% Dm (Tc); sustain time (Ts), as the interval in milliseconds (ms) between 50% of Dm on both the ascending and descending sides of the curve; and relaxation time (Tr), as the interval between 90% and 50% Dm of muscle reaction of the RF, VL, VM and GM, were recorded using TMG. The TMG-derived contraction velocity (Vc) was also calculated by dividing Dm by the sum of Tc and Td [39,42]. In this regard, previous evidence supports the use of Vc as a sensitive marker of acute variations in speed and power performance [39]. All TMG variables used had demonstrated a high intraclass correlation coefficient (ICC) (i.e., 0.86–0.98), as described in previous studies [43,44].

Finally, to assess peripheral fatigue of the lower limb, both legs were individually analyzed, and then the results obtained for each muscle of each leg were pooled, according to García-Manso et al. [29].

### 2.4. Procedures

Both central and peripheral fatigue tests were performed the day before the competition and immediately after it, within 20–30 min of the end of the race, as previously described [38]. Each test lasted no more than 10 min.

Firstly, central fatigue was assessed by measuring HRV. Upon completion of this, peripheral fatigue was assessed using the contractile properties of the RF, VL, VM and GM of both legs. An experimental design scheme is presented in Figure 1.

### 2.5. Statistical Analysis

Statistical analysis of the data was performed using the Jamovi 1.6.16 software (Sydney, Australia). The Shapiro-Wilk test was applied to establish whether the variances of the different variables correspond to a normal and homogeneous distribution. A T-test was performed for repeated samples, or its non-parametric counterpart, when applicable, to detect significant differences before and after the competition (i.e., pre-post) in the following variables: (1) SDNN, Pnn50, RSSMD, SS, and S/PS ratio for central fatigue assessment; and (2) Td, Tc, Ts, Tr, Dm, Vc for peripheral fatigue assessment. Cohen’s *d* was used to measure the effect size (ES) of the parametric meanings, using the small (*d* = 0.2), medium (*d* = 0.5) and large (*d* = 0.8) reference values, as Cohen suggested [45]. In the case of applying a non-parametric test, the ES was determined using a biserial correlation analysis [46]. The confidence interval for the differences was established at 95%. The significant difference for the value of α was established with a value of *p <* 0.05.

## 3. Results

### 3.1. Central Fatigue Assessment: HRV

Table 1 shows the fluctuation of the observed variables, referring to the ANS activity. Resting heart rate (HR) significantly increased after the VK competition (~28%, *p* = 0.01).

The time domain variables showed a significant decrease of the PNS activity after the competition (Table 1). On the contrary, SS, an index related to the SNS activity, significantly increased after the race (Table 1).

Regarding autonomic balance, S/PS ratio increased, denoting a significant predominance of the SNS over the PNS (Table 1).

### 3.2. Peripheral Fatigue Assessment: TMG

Concerning TMG measures, there were several changes in the muscular response after the competition. Table 2 shows the differences between pre- and post-competition values obtained from the analysis of both legs pooled and for each muscle group. Significant differences were found only in TdRF [mean difference: 1.94 (95% CI: 0.41–3.47), t (10) = 2.8310; *p* = 0.018, *d* = 0.85)], TrVM [mean difference: 72.05 (95% CI: 18.01–126.1), t (10) = 2.9707; *p* = 0.014, *d* = 0.89)] and TsGM [mean difference: −139.274 (95% CI: −239.41–39.11), t (10) = −3.09; *p* = 0.011, *d* = 0.93)] when comparing pre- and post-competition values.

Table 3 shows the statistically significant differences obtained from the analysis of each leg individually (lateral symmetry).

## 4. Discussion

The main findings of our investigation were: (1) as expected, there was a significant increase in the SNS activity after the competition, which lasted up to 45–60 min; (2) simultaneously, post-exercise PNS activity was significantly reduced; and (3) time-related variables and Dm levels presented by our runners suggest pre-competition peripheral fatigue.

To the best of our knowledge, this is the first study evaluating central (i.e., ANS performance) and peripheral (i.e., muscle activity) fatigue before and immediately after (i.e., within 20–30 min after finishing) a VK trail running race in recreational trail runners. Although trail running is an emerging topic, the great majority of studies performed in the past few years have focused on metabolic, biomechanical and neuromuscular parameters. However, no studies have simultaneously assessed central and peripheral fatigue via non-invasive methods, especially when uphill trail running is considered (e.g., VK races).

As expected, HRV parameters assessed after the competition showed significantly decreased values when compared to pre-competition levels. However, one interesting point is that when values of the variables related to the modulation of PNS (Table 1) are compared with previous studies [37,47,48], our runners showed lowered values before the VK competition (previous 24–36 h). In this regard, it should be considered that HRV is usually greater in active than sedentary individuals [21,47,49,50]. For instance, trained athletes show higher RMSSD values [21], thus training characteristics might influence HRV time and frequency domain measures [21].

The analyses of the pre-competition HRV time-domain variables (i.e., RMSSD, SDNN and Pnn50) traditionally related to PNS activity [37,51] showed that RMSSD values of our runners (48.45 ± 19.29 ms) would be considered within the average range (i.e., 50th percentile) for the age group when compared with previous studies performed with healthy non-athletes [48]. However, if our results are compared with another investigation carried out in a group of athletes with similar daily activity patterns [47], our runners would be located close to the 25th and 10th percentiles regarding RMSSD and SDNN values respectively. Similarly, Pnn50 values drops to below the 25th percentile when our results are compared with a previous work performed with professional athletes [37]. However, during the present study, a measurement of PNS-related variables was not performed continuously, establishing baseline values and the trend of these before the competition in each participant [52]. Therefore, it was not possible to assess intrasubject PNS activity and to determine a greater or lesser degree of PNS dominance in our runners in the hours prior to competition based only on the HRV measurements performed.

Apart from that, all recorded variables related to time domain underwent a large change after the race (ES *=* 0.89–1.35), indicating a clear downregulation of the PNS (Table 1, time domain variables) and therefore, presumably, an upregulation of the SNS. Thus, it is clear that a VK is a very demanding competition, even if the average speed during the race is low. One of the limitations of this study was the lack of measurements throughout the days after the race, which would allow us to know how much time is needed for the PNS to reach its pre-race levels.

Regarding the pre-competition balance between the SNS and PNS assessed through HRV non-linear measurements, our runners reported high SS values (14.97 ± 7.76) that are beyond the 90th percentile, which is related to a high level of sympathetic stress [37]. Further, the runners of the present study showed a clear disbalance of autonomic activity, reporting an S/PS ratio higher than 0.3 at rest (0.58 ± 0.45), which suggests an excess of the SNS activity or a lack of recovery of the PNS activity [37]. In addition to this, considering the variation of these two variables (i.e., S/PS and SS) before and after the competition, a clear increase in their values can be observed and therefore, a greater dominance of the SNS after the VK. This is in accordance with a lower activity of the PNS measured by time domain variables (Table 1).

Taken together, in addition to the fatigue generated by the VK, both time domain and nonlinear measures may represent a sensitive downward modulation in the PNS of our runners and a lack of autonomic balance (i.e., central fatigue) prior to the race. This interpretation could suggest that a supercompensation status was not attained, which could be linked with fatigue. In this regard, some authors have suggested that recreational endurance runners who do not properly recover between training sessions, or those experiencing psychological stress or autonomic neuromuscular fatigue, are at higher risk of developing the so-called “overtraining syndrome”, which impairs endurance performance and leads to long-term fatigue [53]. Moreover, other studies have suggested that recreational runners try to imitate training practices performed by professional athletes, including high weekly volume (e.g., >70 km), which could lead to some health-related problems (e.g., injuries, overtraining) [54]. With this in mind, we speculate that many recreational runners are usually overtrained and therefore, unable to peak during the competition period. However, the absence of previous works in the field with runners with similar characteristics to those in our study and the lack of continuous pre-race HRV measurement, means it is not possible to draw a conclusion about the degree of stress and fatigue that runners experienced in our study previous to the race.

Regarding peripheral fatigue, despite having analyzed six variables by TMG (i.e., Td, Tc, Ts, Tr, Vc and Dm) in four muscle groups (i.e., RF, VL, VM and GM), there were statistically significant differences only in TdRF (*p* = 0.018; *d* = 0.85), TrVM (*p* = 0.014; *d* = 0.89) and TsGM (*p* = 0.011; *d* = −0.93), when the pooled data of both legs were considered. These variations in Td, Tr and Ts might be connected to metabolic changes in myoplasmic Ca^2+^ [55,56]. Some research in endurance sports suggests that the time-related variables (Td, Tc, Ts, Tr and Vc) tend to decrease after competition, while Dm increases, indicating a reduction of muscular stiffness and an increase of neuromuscular peripheral fatigue [11,29,38,55,57], according to the different muscles evaluated. In this regard, our results are partially in accordance with previous studies, since not all time-related variables showed decreased values after the VK race. For instance, after the race, Tc increased in all the muscles studied, except VL (−1.4%); Ts increased in GM (22.8%); and Vc increased in all the muscles analyzed (RF: 9.2%; VL: 9.2%; VM: 2.9%), except in GM, which decreased (−4.8%). Similarly, Dm increased in all the muscles studied (RF: 8.3%; VL: 7.0%; VM: 3.2%) except in GM, where it decreased (−6.6%). On the other hand, when analyzing each leg individually (i.e., lateral symmetry) we found statistically significant differences in the right leg in TdRF (0.023; ES = 0.94), VcRF (0.032; ES = 0.86), TdVL (0.02; ES = 0.97) and TcVL (0.021; ES = 0.95); and in the left leg in TrVM (0.023; ES = 0.93). These differences between segments may be due to ground surface irregularities, independent of differences in muscle strength, which may predispose the athlete to temporary asymmetric stimuli due to the activity being performed at a given time [58]. Based on our neuromuscular results, and the lack of consistency of previous studies, we speculate that: (1) 20–30 min is a sufficient period for experienced trail runners to recover at neuromuscular level after an intense effort with predominantly concentric contractions (i.e., uphill trail running); and (2) regarding HRV tendency, recreational runners might have already experienced peripheral fatigue before the race. Therefore, the peripheral fatigue generated during the competition did not represent an important stimulus at neuromuscular level. On this point, previous scientific evidence has suggested there is a link between central and peripheral fatigue. Thus, endurance training induces central fatigue adaptations, leading to improved tolerance of peripheral fatigue by the central nervous system [59].

One of the limitations of our study is the small sample size. However, it is important to consider that VK competitions are highly-demanding trail running races with less participation than other endurance running events. The complexity of assessing both central and peripheral fatigue immediately after the race (i.e., in the mountains, within 20–30 min after finishing) makes it difficult to evaluate a large number of participants. Future longitudinal studies are guaranteed for investigating the mechanisms underlying fatigue in endurance trail running events, especially when uphill running (i.e., VK races) is considered. In this regard, it would be of great interest to assess recovery variables over subsequent days after the competition to analyze recovery status evolution.

## 5. Conclusions

The present study demonstrates that a VK race affects the ANS system, downregulating the PNS and upregulating the SNS. However, regarding peripheral fatigue, only small changes in the contractile capacity of specific muscle groups were detected. In addition, the pre-race measurements of HRV could suggest trail runners experienced a lack of recovery or non-functional overreaching before the race. Furthermore, while the neuromuscular stimulus of the competition did not seem to infer a great peripheral fatigue in these athletes, central fatigue significantly increased after the race. Considering the link between both central and peripheral fatigue, training periodization and new tapering strategies are called to play a key role in minimizing pre-competition fatigue status, thus favoring running performance. In this regard, monitoring HRV during the preparation period has been shown to be an effective strategy to avoid pre-competition fatigue.

## Figures and Tables

**Figure 1 ijerph-20-00402-f001:**
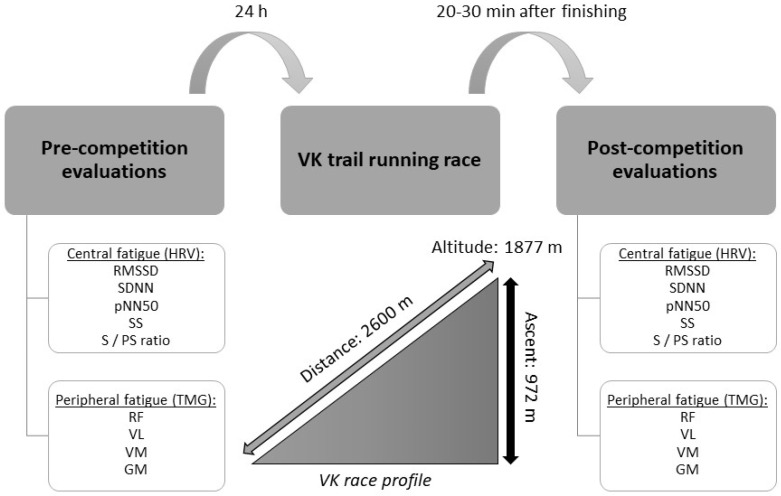
Experimental design scheme. VK: vertical kilometer; HRV: heart rate variability; RMSSD: square root of the mean of the squared differences between successive normal–to–normal intervals; SDNN: standard deviation of normal–to–normal intervals; pNN50: percentage of successive RR intervals that differ by more than 50 ms; SS: stress score; S/PS ratio: ratio between the sympathetic nervous system and the parasympathetic nervous system activity; TMG: tensiomyography; RF: rectus femoris; VL: vastus lateralis; VM: vastus medialis; GM: gastrocnemius medialis.

**Table 1 ijerph-20-00402-t001:** Central fatigue before and after the VK trail running race.

	Pre-VK	Post-VK	*p*-Value	95% CI		Cohen’s *d* ES	95% CI	
	(Mean ± SD)	(Mean ± SD)		Lower	Upper		Lower	Upper
HR (bpm)	56.10 ± 6.96	71.90 ± 11.78	0.01	−26.99	−4.51	−1.17	−2.07	−0.23
Time domain								
SDNN (ms)	58.10 ± 27.06	33.40 ± 18.99	0.02	4.79	44.52	1.04	0.14	1.89
pNN50 (%)	23.71 ± 12.56	8.26 ± 8.94	0.23	3.04	24.49	0.89		
RMSSD (ms)	48.45 ± 19.29	27.21 ± 12.16	0.01	8.13	34.35	1.35	0.35	2.31
Non-linear measurements								
SS (a.u.)	14.97 ± 7.76	28.42 ± 13.66	0.01	−22.31	−4.59	−1.27	−2.20	−0.30
S/PS ratio (a.u.)	0.58 ± 0.45	2.36 ± 2.32	0.02	−3.89	−0.21	−0.94		

HR: Heart Rate; SDNN: standard deviation of normal-to-normal intervals; pNN50: percentage of successive RR intervals that differ by more than 50 ms; RMSSD: square root of the mean of the squared differences between successive normal-to-normal intervals; SS: Stress Score; S/PS ratio: ratio between the sympathetic nervous system and the parasympathetic nervous system activity; VK: vertical kilometer; SD: standard deviation; CI: confidence interval.

**Table 2 ijerph-20-00402-t002:** Peripheral fatigue before and after the VK trail running race for both legs.

Muscle Group	TMG	Pre-VK (Mean ± SD)	Post-VK (Mean ± SD)	Difference	*p*-Value	95% CI		Cohen’s *d* ES	95% CI	
				(%)		Lower	Upper		Lower	Upper
Rectus Femoris (RF)	Td	47.45 ± 2.93	45.34 ± 2.66	−4.7	0.02 *	0.41	3.47	0.85	0.14	1.54
	Tc	57.19 ± 7.09	57.41 ± 8.40	0.4	0.88	−5.0	4.35	−0.05	−0.64	0.55
	Ts	307.43 ± 599.85	143.03 ± 131.92	−114.9	0.17	−97.48	481.47	0.44	−0.19	1.06
	Tr	121.65 ± 230.21	53.69 ± 55.11	−126.6	0.15	−33.34	188.73	0.47	−0.17	1.09
	Dm	14.30 ± 3.43	15.60 ± 3.76	8.3	0.18	−3.10	0.68	−0.43	−1.04	0.20
	Vc	0.27 ± 0.06	0.30 ± 0.06	9.2	0.12	−0.06	0.008	−0.50	−1.13	0.14
Vastus Lateralis (VL)	Td	43.93 ± 2.31	42.81 ± 2.49	−2.6	0.09	−0.19	2.20	0.56	−0.09	1.19
	Tc	45.85 ± 4.99	45.20 ± 5.88	−1.4	0.52	−1.22	2.24	0.19	−0.40	0.79
	Ts	90.84 ± 47.23	89.07 ± 25.18	−2.0	0.81	−28.24	35.47	0.07	−0.52	0.67
	Tr	38.82 ± 39.04	35.05 ± 17.45	−10.8	0.66	−21.23	32.03	0.13	−0.46	0.73
	Dm	10.37 ± 1.82	11.16 ± 1.91	7.0	0.12	−1.68	0.23	−0.50	−1.13	0.13
	Vc	0.23 ± 0.04	0.25 ± 0.04	9.2	0.06	−0.04	0.001	−0.63	−1.28	0.03
Vastus Medialis (VM)	Td	42.83 ± 1.52	42.51 ± 1.01	−0.7	0.49	−0.72	1.38	0.21	−0.39	0.81
	Tc	47.64 ± 4.52	48.15 ± 5.28	1.0	0.41	−1.89	0.83	−0.26	−0.86	0.34
	Ts	411.90 ± 165.18	371.33 ± 59.77	−10.9	0.23	−34.64	127.56	0.38	−0.24	0.99
	Tr	230.80 ± 65.42	151.23 ± 68.48	−52.6	0.01 *	18.01	126.10	0.89	0.17	1.59
	Dm	15.58 ± 3.46	16.09 ± 3.38	3.2	0.24	−1.28	0.36	−0.37	−0.98	0.25
	Vc	0.35 ± 0.09	0.36 ± 0.08	2.9	0.33	−0.03	0.01	−0.31	−0.91	0.30
Gastrocnemius Medialis (GM)	Td	39.94 ± 1.91	37.72 ± 3.82	−5.9	0.10	−0.43	4.12	0.54	−0.10	1.17
	Tc	43.22 ± 6.08	44.10 ± 13.28	2.0	0.66	−8.64	5.73	−0.13	−0.73	0.46
	Ts	480.16 ± 242.51	622.28 ± 353.28	22.8	0.01 *	−239.42	−39.12	−0.93	−1.63	−0.20
	Tr	175.05 ± 129.45	143.02 ± 118.09	−22.4	0.61	−96.87	157.16	0.15	−0.44	0.75
	Dm	5.07 ± 2.17	4.76 ± 2.14	−6.6	0.65	−1.07	1.65	0.14	−0.46	0.73
	Vc	0.12 ± 0.05	0.12 ± 0.05	−4.8	0.71	−0.02	0.04	0.11	−0.48	0.71

VK: vertical kilometer; SD: standard deviation; CI: confidence interval; ES: effect size; TMG: variables derived from the tensiomyography measurements; Td: time delay between the initiation and 10% of maximal radial muscle-belly displacement; Vc: tensiomyography-derived contraction velocity; Tc: contraction time between 10% and 90% of maximal radial muscle-belly displacement; Ts: sustain time, the interval in milliseconds (ms) between 50% of Dm on both the ascending and descending sides of the curve; Tr: relaxation time, the interval between 90% and 50% Dm of muscle reaction. * *p* < 0.05.

**Table 3 ijerph-20-00402-t003:** Peripheral fatigue before and after the VK trail running race for each leg.

Side & Muscle Group	TMG	Pre-VK (Mean ± SD)	Post-VK (Mean ± SD)	*p*-Value	95% CI		Cohen’s *d* ES	95% CI	
					Lower	Upper		Lower	Upper
Right RF	Td	23.74 ± 1.70	22.38 ± 1.95	0.02	0.02	2.46	0.94	0.13	1.71
	Vc	0.14 ± 0.04	0.16 ± 0.036	0.03	−0.04	−0.002	−0.86	−1.62	−0.07
Right VL	Td	22.08 ± 1.48	21.13 ± 1.12	0.02	0.20	1.71	0.97	0.15	1.75
	Tc	23.1 ± 1.83	21.85 ± 2.19	0.02	0.24	2.27	0.95	0.13	1.73
Left VM	Tr	117.4 ± 48.07	67.18 ± 39.76	0.02	8.87	91.55	0.93	0.12	1.71

VK: vertical kilometer; SD: standard deviation; CI: confidence interval; ES: effect size; TMG: variables derived from the tensiomyography measurements; RF: rectus femoris; VL: vastus lateralis; VM: vastus medialis; Td: time delay between the initiation and 10% of maximal radial muscle-belly displacement; Vc: tensiomyography-derived contraction velocity; Tc: contraction time between 10% and 90% of maximal radial muscle-belly displacement; Ts: sustain time, the interval in milliseconds (ms) between 50% of Dm on both the ascending and descending sides of the curve; Tr: relaxation time, the interval between 90% and 50% Dm of muscle reaction.

## Data Availability

The data presented in this study are available on request from the corresponding author. The data are not publicly available due to confidentiality and anonymity of study participants.

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
