# Peer review of "Central and Peripheral Fatigue in Recreational Trail Runners: A Pilot Study"

_ijerph, 2022, doi:10.3390/ijerph20010402_

Round 1

Reviewer 1 Report

Congratulations to the authors for the work. I thought it was an original theme.

Below, I comment on some necessary recommendations to improve the quality of the manuscript:

Abstract. It is recommended to organize the abstract in the sections: introduction, methodology, results and conclusion. It is important to define the study design.

Material and Methods. I recognize that the collection of a sample in such a specific topic is complicated. Was any type of randomization performed to minimize bias?

I see decompensation in sex of the study subjects. How could this be justified?
